# Host-Induced Gene Silencing of the *Aspergillus flavus O*-Methyl Transferase Gene Enhanced Maize Aflatoxin Resistance

**DOI:** 10.3390/toxins17010008

**Published:** 2024-12-27

**Authors:** Olanike Omolehin, Yenjit Raruang, Dongfang Hu, Zhu-Qiang Han, Surassawadee Promyou, Robert L. Brown, Qijian Wei, Kanniah Rajasekaran, Jeffrey W. Cary, Kan Wang, Dan Jeffers, Zhi-Yuan Chen

**Affiliations:** 1Department of Plant Pathology and Crop Physiology, Louisiana State University Agricultural Center, Baton Rouge, LA 70803, USA; nikeomolehin@yahoo.com (O.O.); yraruang76@gmail.com (Y.R.); dhu@agcenter.lsu.edu (D.H.); 2Southern Regional Research Center, Agricultural Research Service, United States Department of Agriculture (USDA/ARS), New Orleans, LA 70124, USA; robert.brown@usda.gov (R.L.B.); qijian.wei@usda.gov (Q.W.); jeffery.cary@usda.gov (J.W.C.); 3Cash Crops Research Institute, Guangxi Academy of Agricultural Sciences, Nanning 530007, China; hanzhuqiang@163.com; 4Faculty of Natural Resources and Agro-Industry, Kasetsart University, Chalermphrakiat Sakonnakhon Province Campus, Sakonnakhon 47000, Thailand; csnsrwd@ku.ac.th; 5Department of Agronomy, Iowa State University, Ames, IA 50011, USA; kanwang@iastate.edu; 6USDA-ARS Corn Host Plant Resistance Research Unit, Mississippi State, Starkville, MS 39762, USA; dan.jeffers@usda.gov

**Keywords:** aflatoxin, *Aspergillus flavus*, *O*-methyl transferase *(omtA)*, *AflP*, host-induced gene silencing, aflatoxin resistance, maize transformation, small RNA, develop transgenic maize lines, RNAi

## Abstract

Maize is one of the major crops that are susceptible to *Aspergillus flavus* infection and subsequent aflatoxin contamination, which poses a serious health threat to humans and domestic animals. Here, an RNA interference (RNAi) approach called Host-Induced Gene Silencing (HIGS) was employed to suppress the *O*-methyl transferase gene (*omtA*, also called *aflP*), a key gene involved in aflatoxin biosynthesis. An RNAi vector carrying part of the *omtA* gene was introduced into the B104 maize line. Among the six transformation events that were positive for containing the *omtA* transgene, OmtA-6 and OmtA-10 were self-pollinated from T1 to T4, and OmtA-7 and OmtA-12 to the T6 generation. These four lines showed at least an 81.3% reduction in aflatoxin accumulation at the T3 generation under laboratory conditions. When screened under field conditions with artificial inoculation, OmtA-7 at T5 and T6 generations and OmtA-10 at T4 generation showed a reduction in aflatoxin contamination between 60% and 91% (*p* < 0.02 to *p* < 0.002). In order to develop commercial maize lines with enhanced aflatoxin resistance, the *omtA* transgene in OmtA-7 was introduced into three elite inbred lines through crossing, and the resulting crosses also exhibited significantly lower aflatoxin accumulation compared to crosses with non-transgenic controls (*p* < 0.04). In addition, high levels of *omtA*-specific small RNAs were only detected in the transgenic kernel and leaf tissues. These results demonstrate that suppression of *omtA* through HIGS can enhance maize resistance to aflatoxin contamination, and this resistance can be transferred to elite backgrounds, providing a viable and practical approach to reduce aflatoxin contamination in maize.

## 1. Introduction

Maize (*Zea mays* L.) is a major staple across the world, but infection by *Aspergillus flavus* and the production of the most potent naturally occurring carcinogenic secondary metabolite known as aflatoxin remain a major health concern for growers and consumers worldwide [1,2]. Although *A. flavus* causes ear rot on maize, it was only considered a minor disease until the discovery of aflatoxin, which led to increased plant breeding and pathology efforts towards reducing *A. flavus* infection and aflatoxin accumulation in maize and other susceptible crops, such as peanuts [3]. This is because aflatoxin is associated with adverse health effects, such as hepatic dysfunction, teratogenic defects, reduced immunity in humans, and stunting and reduced weight in children, particularly in developing countries [4,5]. Occasionally, consumption of highly aflatoxin-contaminated maize products also results in aflatoxicosis and deaths [4,6,7,8]. Due to the health risks associated with aflatoxin, the Food and Drug Administration (FDA) strictly prohibits interstate commerce of grains contaminated with levels of aflatoxin higher than 20 ng/g. The European Union has set even lower limits of 2 ng/g (equivalent to parts per billion, ppb) for aflatoxin B1 in cereals, dried fruits, and tree nuts, and 5 ng/g of aflatoxin B1 in maize and rice that must undergo sorting or other physical treatment before being placed on the market. However, such regulations may negatively impact the economies of countries relying on maize and peanut imports or exports [5,9,10].

Although efforts to reduce aflatoxin accumulation have been intensified, there are no commercial maize hybrids with an acceptable level of resistance to aflatoxin contamination. Currently, the only effective pre-harvest method for aflatoxin management involves the application of biological control agents, such as atoxigenic strains of *A. flavus* NRRL30797, AF36, and NRRL21882. They can reduce aflatoxin contamination in maize and peanuts by up to 94.8% [11,12]. However, the use of these atoxigenic *A. flavus* strains for biological control depends heavily on the availability of resources. In addition, it incurs added costs for farmers. Therefore, other more economical alternative strategies that can protect maize from *A. flavus* infection and aflatoxin contamination are desirable. Efforts in conventional breeding for aflatoxin-resistance in maize have yielded a few aflatoxin resistant inbred lines, such as GT-MAS:gk, MI82, Tex6, Mp420, Mp313E, Mp715, Mp717-Mp719, and TZAR101-TZAR106 [13,14,15,16,17,18,19]. The poor agronomic traits of these lines and the polygenic quantitative nature of aflatoxin resistance have hindered the development of aflatoxin-resistant commercial maize hybrids [20,21].

The identification and characterization of resistance-associated genes and proteins in maize that are differentially expressed during *A. flavus* infection have led to a better understanding of host resistance mechanisms [22,23,24,25]. Some of the resistance-associated genes have been used in genetic engineering to improve crop resistance to *A. flavus*, such as the overexpression of a chitinase or a glucanase gene to enhance aflatoxin resistance in peanuts [26,27]. In addition, a cluster of genes involved in aflatoxin biosynthesis has been identified, and their roles in aflatoxin biosynthesis were demonstrated in *A. flavus* [28,29,30]. Notably, *aflR*, encodes a transcription factor and is a key regulator that modulates the expression of most of the aflatoxin pathway genes [31,32]. Other important genes, *aflB* and *aflA*, are involved in hexanoate formation during the initial steps of aflatoxin biosynthesis [33]. The *aflD*, *aflE*, and *aflF* genes encode the enzymes that catalyze the reactions involved in the conversion of norsolorinic acid (NOR) to averantin during aflatoxin biosynthesis [34,35]. The *aflP* gene (*omtA*), which encodes the *O*-methyltransferase A, is involved in catalyzing the conversion of sterigmatocystin (ST) and DHST to *O*-methylsterigmatocystin (OMST) and dihydro-O-methylsterigmatocystin (DHOMST) during aflatoxin biosynthesis, based on gene disruption and expression studies of the recombinant protein [36,37].

Recent studies have demonstrated in several crops that the reduction of fungal infections and plant diseases can be achieved by suppressing the expression of key pathogen genes via host-expressed double-stranded RNA (dsRNA), which is also referred to as cross-kingdom gene silencing or HIGS [38,39,40,41,42,43,44,45]. For example, the suppression of a Zn II transcription factor (*FOW2)* and class V chitin synthase (*chsV*) genes in the *Fusarium oxysporum* led to reduced vascular wilt disease symptoms in transgenic tomatoes [42]. Several genes involved in regulating deoxynivalenol (DON) biosynthesis and the formation of fungal structures in *Fusarium graminearum* have also been silenced via HIGS, leading to reduced mycotoxin DON production and head blight disease in wheat [43].

This same approach has also been used to successfully mitigate aflatoxin contamination in maize by suppressing *A. flavus aflR*, *aflC*, and *aflM,* or genes involved in fungal growth, such as *alk*, *amy1,* and *p2c* [46,47,48,49,50]. These studies showed a clear reduction in aflatoxin accumulation under laboratory and greenhouse conditions [47,48], and even under repeated field conditions [46,49]. The study by Raruang et al. [46] and Omolehin et al. [49] further demonstrated that the reduced aflatoxin contamination in the transgenic HIGS lines is the result of the presence of high levels of gene-specific small RNAs in the leaf and kernel tissues.

The main objective of the present study was to determine whether suppressing the *omtA* (*aflP*) gene through HIGS can reduce *A. flavus* infection and/or aflatoxin accumulation in maize. The *omtA* gene, unlike previously HIGS-targeted aflatoxin pathway genes, functions at the end of the aflatoxin biosynthesis pathway, which may prove more effective in reducing aflatoxin biosynthesis during *A. flavus* infection when it is suppressed. Therefore, an RNAi-OmtA vector carrying a partial *omtA* gene was constructed according to Chen et al. [51] and introduced into the maize B104 line through Agrobacterium-mediated transformation in late 2012.

The homozygous seeds of the resulting transgenic maize lines were produced and examined for changes in resistance to aflatoxin accumulation under laboratory and field conditions. Under laboratory kernel screening assay (KSA) conditions, aflatoxin reduction ranging between 77% and 93% was observed in the selected transgenic lines. Under artificial field inoculation, two of the four transgenic maize lines (OmtA-7 and OmtA-10) exhibited an average of 85% less aflatoxin production compared to the null control of the same event. These same two lines also exhibited significantly less aflatoxin contamination under a simulated scenario of poor storage conditions. F1 crosses between the transgenic OmtA-7 line and three selected aflatoxin-susceptible elite maize inbred lines also showed enhanced aflatoxin resistance, with a 51.5% to 72.3% reduction in aflatoxin production. RNA sequencing of both transgenic and null controls (leaf and kernel tissues) detected high levels of *omtA*-specific small RNA only in the transgenic lines, which may serve as precursors for silencing the *omtA* during infection, thereby resulting in reduced aflatoxin accumulation. Our findings demonstrate that transgenic maize lines carrying the HIGS *omtA* cassette can significantly reduce aflatoxin contamination in maize. This study also draws closer attention to variations in resistance achievable by HIGS depending on the genes selected for silencing as well as variations among different independent transformation events.

## 2. Results

### 2.1. Construction of the HIGS Vector, Its Transformation into Maize, and Confirmation of the omtA Gene in Trasngenic Maize

The HIGS (pTF102-*omtA*-RNAi) vector (Appendix A) was assembled as described in the Materials and Methods. The final construct was digested with Mfe I, Kpn I, and EcoR V restriction enzymes to verify the correct assembly. Restriction analysis produced two fragments of the expected sizes of 2,447 and 9,085 bp when digested with EcoR V, three fragments of expected sizes of 229, 2,447, and 8,786 bp when digested with Mfe I, and two fragments of the expected sizes of 1,328 and 10,204 bp when digested with Kpn I (Appendix A). This construct will produce a 390-bp *omtA* dsRNA transcript with a 101-bp single-strand loop in the middle once the transcript is processed in the host plant.

This HIGS vector was introduced into the immature embryo of the maize B104 line through *Agrobacterium tumefaciens*-based transformation in September 2012. The transgenic plants, regenerated from independent transformation events, were pollinated with pollen from B104 between April and May of 2013, and mature seeds from 11 independent transformation events were harvested in May and June of 2013. Based on PCR analysis of the genomic DNA isolated from T0 transgenic leaf tissues from each event received from the Iowa State University (ISU) Plant Transformation Facility, six (OmtA-6, 7, 8, 10, 11, and 12) were confirmed positive for the *omtA* transgene (Figure 1A). Among them, the OmtA-6 event showed the highest target gene expression, followed by OmtA-7, OmtA-10, and OmtA-12 based on qRT-PCR analysis of RNA samples prepared from T0 plant leaf tissues (Figure 1B). These four events were selected for further investigation in the following studies.

### 2.2. Variation in Zygosity and Gene Copy Number

Three (OmtA-6, OmtA-7, and OmtA-12) of the four events mentioned above were believed to have a single integration of the *omtA* gene in the maize genome, based on the chi-square analysis of the number of T2 generation transgenic and non-transgenic seedlings determined through PCR genotyping (Table 1). OmtA-10 exceeded the critical value for one integration, suggesting the presence of possible multiple integrations (Table 1). Further analysis using the real-time PCR revealed twice the zygosity of the T0 in all purported T3 homozygous samples (Table 2). The genomic DNA isolated from the leaf tissues of all six of the T0 events that tested positive for transformation and all four of the T4 events were quantified for *omtA* copy number relative to the single copy reference gene alcohol, dehydrogenase gene (*adh1*), using the droplet digital PCR (ddPCR). The target gene copy number based on ddPCR analysis of the ratio of *omtA/adh1* in the genomic DNA of T0 leaf samples, confirmed a single copy of the transgene for five of the six events analyzed (Table 3). The other event (OmtA-6) has four copies of the transgene (Table 3). The ddPCR also confirmed that all four of the T4 events are homozygous for the *omtA* transgene.

### 2.3. Aflatoxin B1 Production in the Transgenic Kernels at T1 and T3 Generations

Five *omtA*-positive transgenic T1 events with sufficient seeds were analyzed for differences in aflatoxin production through the in vitro kernel screening assay (KSA) (Appendix A). The transgenic T1 kernels from four (OmtA-6, 7, 10, and 12) out of the five events exhibited lower mean aflatoxin production than those without the *omtA* transgene, although the difference is not statistically significant except for OmtA-7 and OmtA-12, which were at borderline significance with *p*-values ranging from 0.056 to 0.063 (Appendix A).

The seeds from these four events were increased in the field by self-pollination to T2 in 2015, and again to T3 in 2016. When the T3 generation homozygous and null kernels from the OmtA-6, OmtA-7, OmtA-10, and OmtA-12 events were evaluated again using KSA, a significant reduction in aflatoxin accumulation ranging from 81.3% (OmtA-12) to 98.8% (OmtA-7) was observed between homozygous transgenic and null controls in all four events (with *p*-values ranging from <0.007 to <0.02) (Figure 2).

### 2.4. Phenotypic Variations of Homozygous Transgenic and Null Control Plants at T3 Generation and Ears at T4 Generation

The phenotypic traits of T3 plants and T4 ears were evaluated for differences between the homozygous transgenic and their null controls (Table 4). The height of the homozygous transgenic OmtA-RNAi lines and their null controls ranged from 244 to 307 cm. The cob length among the homozygous lines and the null controls ranged between 13.4 and 16.2 cm on average (Table 4). The 100-kernel weight showed similar variations, ranging from 21.1 to 27.8 g for both homozygous lines and their null controls. A significant time delay in silking and an increase in kernel weight were observed for the homozygous transgenic OmtA-7 line compared to its null control. In contrast, a significantly lower kernel weight was observed for the homozygous transgenic OmtA-12 compared to its null control. The transgenic OmtA-6 also exhibited significant delay in silking compared to B104. However, the phenotypic variations observed in these events did not appear to be associated with the presence of the transgene since both OmtA-6 null and OmtA-12 null also significantly varied in plant height compared to the B104 control. Additionally, no clear visual detrimental morphological changes were observed in the transgenic plants compared to the null or B104 plants (Table 4, Figure 3).

### 2.5. Aflatoxin Production in the Kernels of OmtA-6, OmtA-7, OmtA-10, and OmtA-12 Events Under Field Conditions

The ears of OmtA-7 and OmtA-12 plants at the T4 generation in 2017 suffered severe *Fusarium* spp. Infection, resulting in relatively low aflatoxin values under our field inoculation conditions were relatively low (about 1000 ng/g or less). There was no significant difference in aflatoxin levels between the kernels from OmtA-7 or OmtA-12 homozygous transgenic lines and the kernels from their corresponding null controls or B104 (*p* = 0.49, and 0.65, respectively; number of ears = 5 to 12) (Appendix A).

The OmtA-7 and OmtA-12 lines were evaluated again under field conditions at the T5 (2018) and T6 (2019) generations. A significant reduction (up to 58.6%) in aflatoxin accumulation was observed in the T5 homozygous OmtA-7 transgenic line in comparison to the null control (*p* < 0.027) (Figure 4A). Although 47.6% less aflatoxin accumulation was observed for OmtA-12 homozygous line, the difference in aflatoxin levels between the OmtA-12 and its null control was only borderline significant (*p* < 0.058) (Figure 4A). In the T6 generation, OmtA-7 again a showed significant (*p* < 0.002) reduction in aflatoxin accumulation (64.8% reduction) compared to OmtA-12, which showed only a 24.2% reduction when compared to the null control (Figure 4B). This difference was not statistically significant (*p* < 0.36; number of ears = 7). Two additional events (OmtA-6 and OmtA-10 at T4 generation) were also evaluated in 2019. OmtA-10 showed a significantly lower aflatoxin production (90.4% reduction) compared to its corresponding null control (*p* < 0.004; number of ears = 13) (Figure 4C). However, only a 33.1% reduction in aflatoxin production was observed in the homozygous transgenic OmtA-6 compared to its null control, which was not statistically significant (*p* < 0.59; number of ears = 7) (Figure 4C).

### 2.6. Transgenic Lines Produced Significantly Less Aflatoxin Under Simulated Storage Conditions

To determine whether the OmtA construct can also provide post-harvest protection to maize against aflatoxin contamination during storage, harvested, naturally infected mature kernels from the T6 generation of OmtA-7, 12 and T4 generation of OmtA-6, 10 were surface sterilized and incubated for 7 days under KSA conditions without inoculation. Kernels from these lines produced barely detectable levels (less than 1 ng/g) of aflatoxin B1, except for OmtA-12 null and OmtA-12 homozygous transgenic lines, which had a detected levels of aflatoxin B1 of 12.9 ng/g and 1.1 ng/g (91.3% reduction compared to the null), respectively (Figure 5A). With inoculation, higher aflatoxin accumulation was observed for all the lines examined. There was no significant difference in the levels of aflatoxin between homozygous lines and their null controls for OmtA-6 and OmtA-12 (Figure 5B). However, significantly less aflatoxin production was observed in the homozygous OmtA-7 and OmtA-10 compared to their null controls (*p* < 0.013), with an average reduction in aflatoxin of about 95.0% and 97.1%, respectively (Figure 5B).

### 2.7. The Presence of the OmtA Transgene Conferred Aflatoxin Resistance in the F1 Crosses with Elite Lines Under Field Conditions

To verify whether the significantly lower aflatoxin production in the transgenic in the OmtA-7 event is due to the presence of the transgene, three elite inbred lines (PHN46, LH195, and PHG39) were crossed with pollen from the homozygous OmtA-7 and its null plants. All of the single-cross hybrids and the homozygous OmtA-7 produced significantly less aflatoxin than the crosses between the elite lines and the null control (*p* < 0.02–0.041) (Figure 6). On average, a 51.5% to 72.3% reduction in aflatoxin accumulation was observed in the OmtA-hybrids compared to the selfed elite lines or the crosses of the elite lines with the null controls.

### 2.8. Small RNA Expression in Homozygous OmtA-RNAi Lines Compared to Null and Negative Controls

To assess whether the improved aflatoxin resistance in the OmtA transgenic lines compared to the null controls was the result of *omtA*-specific small RNAs arising from the long dsRNA transcribed from the OmtA-RNAi cassette in the transgenic line, small RNA libraries generated from T3 leaf and T4 kernel tissues were sequenced and analyzed (Table 5). In the T3 generation transgenic leaf tissues of OmtA-7 line, *omtA*-specific small RNAs accounted for about 20.5% of total small RNAs aligned to the *A. flavus* genome, whereas only 0.04% of *omtA*-specific small RNA were detected in B104 control leaf tissue (Table 5). The total number of small RNA reads in the T4 OmtA-7 homozygous transgenic kernel tissues that were aligned to *A. flavus* was 51,791, of which 12,858 (24.8%) were aligned to the *omtA* gene. In the OmtA-12 homozygous line, 78,161 small RNA reads were aligned to *A. flavus* and 11,658 (14.9%) of the reads were aligned to the *omtA* gene. Only several to fewer than a hundred small RNAs from libraries of transgenic plants were aligned to other non-target *A. flavus* genes, similar to the number of reads of small RNAs from the null and B104 plants that were aligned to *omtA*, which ranged from 13 to 28 counts (Table 5). Among the small RNAs aligned to the *A. flavus* genome, 21, 22, and 24 nt in length in the leaf or kernel of the transgenic OmtA-7 line were the most abundant (Figure 7A). A high percentage of the sequenced small RNAs came from the antisense strand, especially for OmtA-7 (Figure 7B), although both *omtA*-specific sense and anti-sense small RNAs were observed. The small RNA profiling also showed that most of the small RNAs were produced from a few hotspots within the 380 bp *omtA* transgene region that was transformed into maize (Figure 7C).

## 3. Discussion

The genetic basis of aflatoxin production during *Aspergillus flavus* infection of major susceptible crops such as maize, has been attributed to the presence of as many as 25 aflatoxin biosynthetic pathway genes clustered within a 75-kb region of the fungal genome [28]. Developing pre-harvest resistance to aflatoxin accumulation in maize through the conventional breeding has been very challenging and time-consuming. Therefore, alternative approaches, such as using HIGS to suppress the expression of the aflatoxin biosynthetic pathway genes, such as *aflR*, *aflC,* or *aflM*, to reduce aflatoxin contamination in maize have been attempted in several recent studies [46,47,48,52].

In the present study, a different aflatoxin pathway gene (*omtA*/*aflP*) was targeted for suppression through HIGS by transforming maize with the OmtA-RNAi construct. Two methods were used to assess the transgene copy numbers in each of the transformation events: genotyping of T2 seedlings and performing χ^2^ analysis and ddPCR. For OmtA-10, the Chi-square (χ^2^) analysis of its T2 seedlings indicates the presence of possibly more than one integration of the target gene, while the ddPCR confirmed that it contains only one copy of the target gene. These apparently conflicting results are possibly due to the smaller sample size used in our χ^2^ analysis since the accuracy of ddPCR has been well documented [53,54]. The presence of the transgene in the transgenic lines did not seem to adversely affect the phenotypic characteristics of the transgenic plants based on the preliminary assessment of differences in the plant height and cob length of the transgenic lines compared to the null controls (Figure 3; Table 4), indicating that there is no suppression of off-target genes related to plant growth or development.

To assess the aflatoxin resistance of the resulting transgenic maize lines, two different methods were used in this study: one is the laboratory based KSA assay using mature kernels, and the other is to inoculate immature maize kernels in the field and assay their aflatoxin levels at maturity. The former is fast, with results that can be obtained in about two weeks, and the results are often very consistent. The latter method takes about 6–8 weeks from the time of inoculation to obtain results. The data are often highly variable, but this method reflects more closely to how these lines will perform in the field. Overall, the results obtained from these two methods are consistent in our studies. Sometimes, the same transformation events performed differently when comparing the aflatoxin data from field-inoculated studies to those obtained in KSA. Several factors contribute to this difference. One is that the KSA involves incubating mature kernels for seven days under ideal conditions for *A. flavus* infection and toxin production, with no competition or suppression by other microorganisms, which have been reported to affect aflatoxin production [55,56,57,58]. Second, mature kernels may lack the expression of certain disease or resistance-associated genes that are expressed in immature kernels, such as the expression of the *chitinase* gene on the maize pericarp [59,60]. The third factor is the method of inoculation. For KSA, kernels were inoculated by dipping them into inoculum for one min. However, this method is not possible for inoculating immature kernels on the ears. Instead, the standard side-needle inoculation method was used in field inoculation studies. This involves wounding developing maize kernels and can simulate kernel injury (wounding) caused by insect vectors under field conditions. This method also circumvents the protections provided by physical barriers, such as the husk covers and kernel pericarp. Additionally, environmental factors, such as temperature and moisture [61,62,63], contribute to the variations in aflatoxin data from field studies. All these differences highlight the importance and necessity of evaluating these transgenic lines under field conditions in order to reliably assess their aflatoxin resistance. This is also the reason why only the intact kernels surrounding the needle-wounded ones were collected for aflatoxin analysis in our field evaluation of T4–T6 transgenic lines.

The initial assessment of the aflatoxin resistance in the T1 generation showed lower levels of aflatoxin production, although not statistically significant, in the transgenic kernels compared to the null kernels of the same events (Appendix A). Subsequent laboratory KSA reassessment using homozygous kernels in the T3 generation observed a highly significant reduction in aflatoxin accumulation (ranging from 81.3% to 98.8%) in the transgenic compared to their null controls in the OmtA-6, 7, 10, and 12 events (Figure 2). However, only the transgenic OmtA-7 and OmtA-10 lines demonstrated the most aflatoxin resistance under repeated field evaluations, with consistent and significant suppression of aflatoxin production, ranging from 58.6% to 90.4%.

The lack of consistent reduction in aflatoxin production in the OmtA-6 may be due to the presence of multiple copies of the *omtA* gene, based on droplet digital PCR analysis (Table 3), which may have negatively influenced the efficacy of the target gene silencing [64,65,66]. This may also explain why a significantly higher level of *omtA* expression was detected in OmtA-6. Moreover, the copy number, the variation in the location of the transgene integration among different transformation events could be another contributing factor [67,68,69]. However, the multiple copies of the *omtA* transgene in OmtA-6 appeared to have been integrated at a single location, according to genotyping and Chi-square analysis. The reason for the lack of efficacy in reducing aflatoxin contamination in OmtA-12 remains unclear since it has one copy of the transgene, similar to OmtA-10.

The levels of aflatoxin observed in the inoculated homozygous transgenic OmtA-7 or OmtA-10 lines and in F1 crosses with elite inbred lines were in the range of thousands of ng/g, which is much higher than the 20 ng/g, upper limit of aflatoxin in grains intended for human consumption, as set by the FDA. This is partially due to the high inoculum concentration used in our study. An initial *A. flavus* inoculum concentration of 4 × 10^6^ spores/mL was used for the laboratory study and field inoculations, which consistently yield very high levels of aflatoxin (up to 120,000 ng/g). This concentration is thousands of times higher than what plants may encounter under natural infection conditions, which may overwhelm the resistance conferred by the suppressive small RNAs and diminish our ability to distinguish the performance of the transgenic and null kernels across all events under field conditions. Therefore, the inoculum concentration was reduced to 1 × 10^5^ spores/mL in the subsequent KSA, and field inoculations conducted in 2018 and 2019. Another reason for the high levels of aflatoxin in our study is how the kernels were collected for aflatoxin analysis. In this study, only the kernels from around the inoculation site was harvested for aflatoxin extraction. Aflatoxin analysis conducted by breeders or at commercial elevators uses bulk samples on the scale of kilograms, and their toxin data are usually in the range of a few hundred ng/g or lower [70]. When we used half of the inoculated ears instead of only the 16 kernels surrounding the four inoculation sites, the level of aflatoxin detected was reduced from hundreds of thousands ng/g to a thousand ng/g or less in most samples [71]. Under natural infection conditions, even with 7 days of incubation, the aflatoxin levels were 12.9 ng/g or lower for both transgenic and null controls (Figure 5A). Therefore, HIGS targeting the *omtA* gene has a great potential to reduce aflatoxin contamination not only in the field, but also to protect the harvested grains during storage.

In an effort to develop aflatoxin-resistant maize lines in an elite background, this study further confirmed that the transgene enhanced aflatoxin resistance in the elite background after crossing with the homozygous transgenic OmtA-7 line, resulting in up to a 72.3% reduction in aflatoxin accumulation in the F1 hybrids (Figure 6). Repeating the backcrossing of these F1 crosses with their corresponding elite parental lines can lead to the development of aflatoxin-resistant commercial maize inbred lines with other desirable agronomic traits.

Our small RNA sequencing analysis confirmed the production of high levels of *omtA*-specific small RNA among all the small RNAs aligned to *A. flavus* in our homozygous transgenic kernels of OmtA-7 (24.8%) and OmtA-12 (14.9%), compared to less than 0.07% in their null controls. This higher abundance of *omtA*-specific small RNA detected in the OmtA-7 kernel tissue, compared to OmtA-12, may be one of the reasons for why OmtA-7 consistently performs better in reducing aflatoxin contamination in both the lab and field evaluations than OmtA-12. However, how the siRNAs travel within the plants and from host plants to pathogens is still poorly understood. It appears that the short-distance cell-to-cell movement of silencing signals occurs through plasmodesmata, while the long-distance movement takes place through the phloem system, including the companion cells [72,73]. The phloem small RNA-binding protein 1 (PSRB1) and small RNA-binding protein 1 (SRBP1) isolated from pumpkin can bind specifically to 24 nucleotide (nt) sRNAs and mediate their cell-to-cell movement [74]. Recent studies further indicated that extracellular vesicles play an important role in small RNA trafficking between hosts and pathogens [45,75]. In addition, gene-specific small RNAs of 21 nt and 24 nt have been reported to play critical roles in post-transcriptional and transcriptional gene silencing, respectively [76,77,78]. In our small RNA sequencing, high levels of 21nt and 24 nt *omtA*-specific small RNAs were detected in transgenic OmtA-7 T3 generation leaves and T4 generation kernel tissues.

It is evident that the HIGS approach has several of advantages over biocontrol in mitigating aflatoxin contamination: it does not incur additional costs to farmers during the production, and it can provide protection against post-harvest aflatoxin contamination during storage under suboptimal storage conditions. Future studies may focus on elucidating the pathway involved in dsRNA trafficking between the *A. flavus* pathogen and maize, especially related to the silencing of the *omtA* gene, such as evaluating changes in the level of the *omtA* protein and the expression of other toxin pathway genes as a result of suppression of *omtA* by the HIGS. Other studies could improve maize resistance to aflatoxin accumulation by suppressing multiple aflatoxin pathway and *A. flavus* virulence genes in a single RNAi vector. Therefore, there is still room for further improvement of this HIGS strategy to achieve a successful management of aflatoxin contamination in maize and other susceptible crops to a level lower than the limits set by the US and EU under natural infection conditions.

## 4. Conclusions

The present study demonstrates that silencing the *omtA* gene can be used to reduce aflatoxin contamination in maize through HIGS. Significantly higher levels both of *omtA*-specific small RNAs were detected in the leaf and kernel tissues of transgenic OmtA-7 and OmtA-10 lines, indicating that the reduced aflatoxin contamination in these homozygous transgenic kernels is due to the suppression of *omtA* expression through HIGS.

## 5. Materials and Methods

### 5.1. Construction and Maize Transformation of an RNAi Vector to Target the A. flavus omtA Gene

A Gateway-based vector previously constructed by Chen et al. [51] was used to insert the gene silencing cassette. The sequence of the *omtA* (*aflP*) target gene (accession #XM_002379891) was obtained from *A. flavus* strain AF13. The 5′ and 3′ arms of the *omtA* coding region were amplified through PCR with primers containing homologous recombination sites at the 5′ end of the gene-specific primers (Appendix A), and cloned into pDONR P4-P1R and pDONR P2RP3, respectively, as previously described [49]. The resulting vectors, named pENTR-L4-5′ *omtA*-R1 and pENTR-R2-3′ *omtA*-L3, were sequenced to confirm that the desired sequence was obtained. The homologous recombination to produce the final pBS-d35S-attB4-5′ *omtA*-attB1-PR10 intron-CmR-attB2-3′ *omtA*-attB3 was performed as previously described [51]. The resulting construct was verified through digestions with EcoR V and EcoR I/Kpn I restriction enzymes. The silencing cassette containing d35S-attB4-5′ omt-attB1-PR10 intron-CmR-attB2-3′ omt-attB3 was recovered through double digestion with EcoR I and Sac I and ligated into the corresponding sites of pTF102, as described [51,79]. The resulting maize RNAi transformation vector (pTF102-d35S-Omt RNAi) (Appendix A) was digested with restriction enzymes and sequenced to verify that the vector was assembled as designed.

The above pTF102-d35S-*omtA*-RNAi construct was transformed into the *A. tumefaciens* strain EHA101, which was introduced into the immature embryos of the maize B104 maize line at the Plant Transformation Facility (PTF), Iowa State University (ISU), as described by Raji et al. [80]. Stable calli selected from the bialaphos-containing media were used to regenerate transgenic seedlings, which were pollinated with pollen from B104 between April and May of 2013, and ears were harvested in May and June of 2013. Mature ears from 11 independent events were received in July of 2013 (Appendix A).

### 5.2. Confirmation of Transformation and Target Gene Expression

Genomic DNA and total RNA were isolated from T0 leaf tissues using the modified CTAB method [81], and the RNAeasy Plant Mini Kit (Qiagen, Hilden, Germany), respectively, and quantified using a NanoDrop ND-1000 Spectrophotometer (Thermo Scientific, Wilmington, DE, USA). The diluted genomic DNA (50 ng/µL) was used as a template for PCR with *omtA*-specific primers (Forward primer: CGA CTT GCT TGG GTC CAT and Reverse primer: AAG ATC GGG CAT AAT CAT TTC) (Appendix A) to confirm the presence of the target gene in all events, as described [49]. cDNA was synthesized from 250 ng of the total RNA isolated above using the TaqMan reverse transcription reagents (Applied Biosystems, Foster City, CA, USA) by following the manufacturer’s protocol and was used as a template to quantify the relative expression level of *omtA* in the T0 leaf tissue through qRT-PCR. The amplification efficiency of all the primers used in this study was determined through serial dilution. Five-fold serial dilutions of cDNA or genomic DNA from 100 ng to 10 pg were used as templates in real-time PCR to determine the amplification efficiency of the primers. The qRT-PCR was performed using the Bio-Rad CFX Connect TM Real-Time System (Bio-Rad, Hercules, CA, USA) with *omtA*-specific forward (CGG AGT TGA GGA CAC TGA TAA A) and reverse primers (CAG CAT CGG GAT AGT CAT GTA G) and a TaqMan probe (FAM/ATC GGA GAA/ZEN/GAT AGA CAT CGG CGC/3IABkFQ) (Appendix A) under the same conditions as previously described [49], except that the annealing and elongation were at 54.3 °C for 30 s. The 18S rRNA was used as an internal control, and its expression level was quantified and used to normalize the level of *omtA* expression. Four positive transformation events were used in further evaluation studies.

### 5.3. Increase the Selected Transgenic Lines and Produce Crosses with Elite Inbred Maize Lines

The T1 seeds from four events (OmtA-6, OmtA-7, OmtA-10 and OmtA-12) that exhibited high to medium levels of *omtA* expression were planted in the field at the Burden Farm Research Station in Baton Rouge, Louisiana, and self-pollinated to produce T2 seeds in 2015. This process was repeated 4 more times between 2016 and 2019 to bring OmtA-6 and OmtA-10 to the T4 generation and OmtA-7 and OmtA-12 to the T6 generation. In addition, the elite inbred lines LH195, PHN46, and PHG39 were pollinated with pollen from homozygous OmtA-7 and null-7 plants to incorporate the *omtA* transgene into an elite maize background, producing F1 hybrids in 2018. Information on the number of seeds used, seedlings planted, plants pollinated, ears inoculated, and harvested from T3 to T6 and for the production of F1 crosses is provided in Appendix A. Additionly, 10 to 12 transgenic maize plants from each of the four transformation events were evaluated for differences in phenotypic characteristics, such as plant height from the base of the plant to the flag leaves (cm), dates to 50% tasseling and silking, mature cob length, and 100-kernel weight (g).

### 5.4. Genotyping, Zygosity, and Transgene Copy Number Determination

The seedlings at the T2 generation were genotyped through traditional PCR to confirm the presence or absence of the target gene *omtA,* as described previously [49]. Segregating maize seedlings without the *omtA* gene were referred to as “null control”. Seeds developed from these plants, along with the B104 non-transgenic parental line, were included as negative controls in all field and laboratory studies. The segregating maize seedlings that are positive (either heterozygous or homozygous) for the *omtA* gene were termed “transgenic plants”. For single copy/gene integration, the expected ratio of transgenic vs. null is 3:1. The chi-square analysis was performed to estimate whether a particular transformation event contains one, two or more copies of the *omtA* gene [49].

The zygosity of *omtA* in the seedlings from the T3 generation was determined through qPCR by comparing the ratio of *omtA* to the internal single-copy alcohol dehydrogenase (*adh1*) gene to that in the heterozygous T0 seedlings, as described by Omolehin et al. [49]. Due to the presence of two sets of *omtA* primer binding sites in the 5′ and 3′ arms of the HIGS cassette in the transgenic lines, the expected Z value in the heterozygous T0 seedlings is 1, while the homozygous T3 seedlings are expected to have a calculated Z value close to 2.

To accurately determine the transgene copy number in the above four events, genomic DNA isolated from T4 leaf tissues and from T0 leaf tissues of these same four events, plus OmtA-8 and OmtA-11, were analyzed through droplet digital PCR (ddPCR) at the Interdisciplinary Center for Biotechnology Research (ICBR), University of Florida (Gainesville, FL, USA). Its reliability in determining gene copy numbers has been well documented [53,54,82]. The ddPCR, with primers specific to the *omtA* transgene and maize *adh1* gene (Appendix A), was performed in a C1000 Touch thermal cycler (Bio-Rad, Hercules, CA, USA) and analyzed in a QX200 droplet reader (Bio-Rad) as previously described [49].

### 5.5. Evaluate Transgenic Maize Lines for Changes in Aflatoxin Accumulation Under Laboratory and Field Conditions

Twenty kernels from homozygous OmtA-6, OmtA-7, OmtA-10, and OmtA-12 events at the T3 generation, as well as their non-transgenic controls (null and B104) were evaluated for their resistance to *A. flavus* infection and aflatoxin contamination using the kernel screening assay (KSA) after first surface-sterilizing the kernels by soaking the kernels with 70% ethanol for 5 min, followed by rinsing them with sterile water 3 times to remove any residual ethanol, as described by Brown et al. [18]. For T3 kernels, 20 kernels from each of the homozygous and null lines were used in the KSA. Sterilized and dried seeds were inoculated by dipping in 10 mL of 4 × 10^6^ conidia/mL (in 0.01% SDS solution) of freshly inoculum prepared from a 7-day old *A. flavus* AF13 (ATCC 96044, SRRC 1273) culture on PDA plates. Each inoculated kernel was placed in a petri dish measuring 35 × 10 mm in dimension and incubated at 30 °C under 100% humidity. After 7 days of incubation, the kernels were dried at 65 °C for three days and ground individually using a coffee grinder (Mr. Coffee^®^, Boca Raton, FL, USA).

The aflatoxin resistance of OmtA-7 and OmtA-12 lines was also evaluated under field conditions through artificial inoculations from 2017 (T4) to 2019 (T6). Seven to fourteen immature ears from each line were inoculated at 4 sites/ear in the middle of the ear 14 days after pollination using an Indico tree-marking gun (Forestry Suppliers, Jackson, MS, USA) with a 15-gauge needle. Each ear was inoculated with 2 mL of *A. flavus* AF13 spore suspension according to Williams et al. [59]. The inoculum concentration used in 2017 was 4 × 10^6^ conidia/mL, which was reduced to 1 × 10^5^ conidia/mL in 2018 and 2019 to better simulate a more natural infection. Inoculated ears were harvested at maturity, and the four kernels surrounding each needle injection site were recovered and ground as one sample. Furthermore, the T4 generation kernels from OmtA-6 and OmtA-10 were also evaluated in 2019 under the same field conditions. Aflatoxin analysis was carried out using 7–14 ears with 3–4 samples per ear for each of homozygous, and null line.

For the crosses between elite inbred lines and OmtA-7, 5–15 immature ears were inoculated as above. At maturity, kernels from half of each ear with four inoculation points were recovered and ground as one sample using a GRINDOMIX knife mill GM 200 milling machine, (Retsch USA, Newtown, PA, USA) at a speed of 50 Hz for 10 secs. Three subsamples of 2 g each per ear were used for aflatoxin extraction and analysis. The analysis was conducted using 15–33 subsamples from 5 ears (for null crosses) to 10 (for crosses with the transgene) or 11 ears (for selfed elite parent).

### 5.6. Extraction and Quantification of Aflatoxin

Extraction of aflatoxin from pre-weighed ground kernel samples was performed in 20 mL of 80% methanol in 50-mL flasks as previously described [49]. The extracted aflatoxin in methanol was filtered through No. 1 Waterman filter paper and an alumina silica (CAT 1344-28-1, Fisher Chemical, Switzerland) filled column according to Sobolev and Dorner [83]. The final filtrate was collected into properly labeled vials for aflatoxin quantification.

Aflatoxin was quantified using a High-Performance Liquid Chromatography (HPLC) as described in Sweany et al. [84,85]. Ten μL of each extracted aflatoxin sample in methanol was then injected into a Waters e2695 HPLC (Waters Corporation, Milford, MA, USA) with a reversed-phase Nova-Pak C18 4 µm 3.9 × 150 mm column. The mobile phase was 37.5% methanol and 62.5% water at a 0.8 mL/min flow rate, and the running conditions were as previously described by Raruang et al. [71]. Aflatoxin standard was purchased from Sigma Aldrich (St. Louis, MO, USA) was serially diluted to 1, 5, 10, 50, 100, 500, and 1000 ng/g (ng/mL) and used for standard curve construction. The AFB1 was quantified based on its peak area in the chromatograph compared to those in the standard curve using the included Empower 3 software (Appendix A).

### 5.7. Isolation of Total RNA, Construction and Sequencing of Small RNA Libraries, and Bioinformatics Analysis to Detect Gene-Specific omtA-Small RNA

Total RNA was isolated from T3 leaf tissues of OmtA-7 and the B104 non-transgenic lines, as well as from immature maize kernel tissues of T4 generation of OmtA-7 and OmtA-12 homozygous lines and their null controls 14 days after self-pollination. Sample grinding, RNA extraction and quantification, and indexed small RNA library construction followed the instructions described by Raruang et al. [46]. The indexed small RNA libraries were sequenced, and bioinformatics analysis was performed as previously described [49].

### 5.8. Statistical Analyses

SAS version 9.4 (Statistical Analysis System, SAS Institute, Cary, NC, USA) was used for all the statistical analyses in this study. Analyses of variance (ANOVAs) were calculated using Proc Mixed. Post-hoc comparison of means was calculated using Tukey’s LSD means [86]. All the toxin data were used directly for SAS analysis without transformation except for those from OmtA-7 crosses with three elite inbred lines, which were log-transformed before the analysis to equalize the variation. Significance in this study was defined by a confidence interval ≥95% (α = 0.05). Means or bars followed by the same letters were not significantly different from each other.

## Figures and Tables

**Figure 1 toxins-17-00008-f001:**
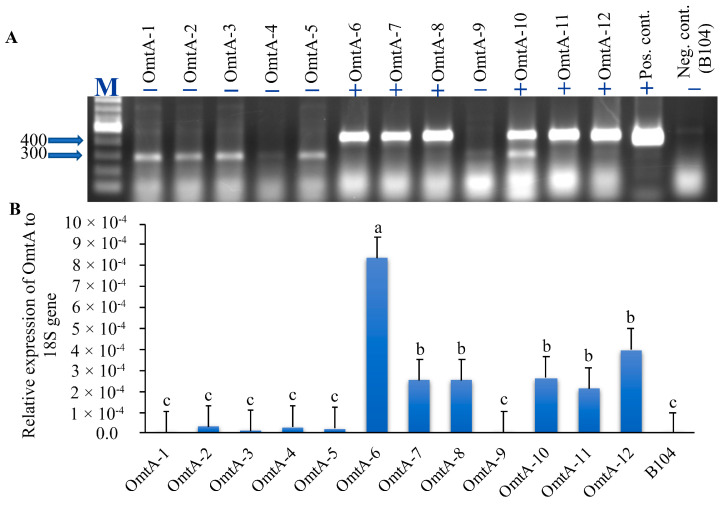
(**A**). Polymerase Chain Reaction (PCR) to confirm the presence (+) or absence (−) of the *omtA* target gene in the leaf tissue collected from various transformation events. The plasmid DNA containing the *omtA* gene and the genomic DNA from B104 leaf tissue were used as templates for the positive and negative controls, respectively. (**B**). Real-time PCR quantification of transcript levels of *omtA* in the leaf tissue of different transformation events at T0. The expression level was normalized to that of 18S rRNA. OmtA-1 to OmtA-5 and OmtA-9 are negative for the transgene. Bars labeled with the letter “a” showed the highest expression of the *omtA* gene, while those labeled with the letter “c” showed the lowest expression of the *omtA* gene; the expression levels in the events with the same letters were not significantly different at *p* ≤ 0.05.

**Figure 2 toxins-17-00008-f002:**
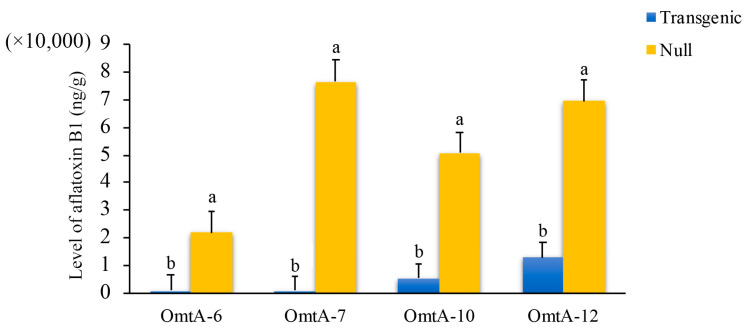
Aflatoxin accumulation in T3 generation *omt-A* homozygous transgenic lines and null controls at under laboratory kernel screening assay (KSA) conditions. Twenty kernels per line were inoculated with *Aspergillus flavus* conidia suspension (4 × 10^6^ conidia/mL) and incubated at 30 °C for 7 days before the kernels were dried, ground, and extracted for aflatoxin. Bars with different letters are significantly different at *p* < 0.05.

**Figure 3 toxins-17-00008-f003:**
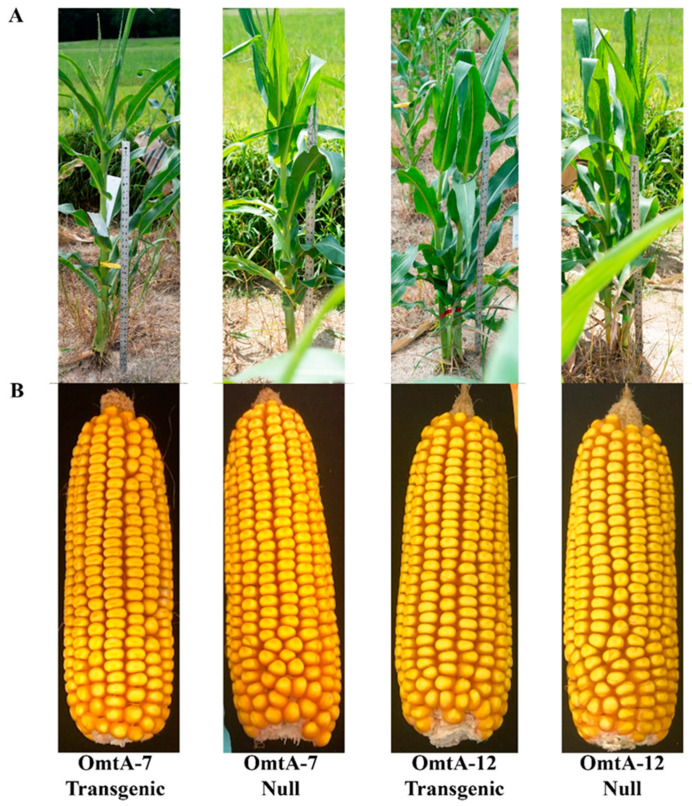
Representative appearance of plant architecture and height (**A**), and ear length and kernel set (**B**) of OmtA-RNAi transgenic and null T3 plants at flowering, and T4 ears at harvest.

**Figure 4 toxins-17-00008-f004:**
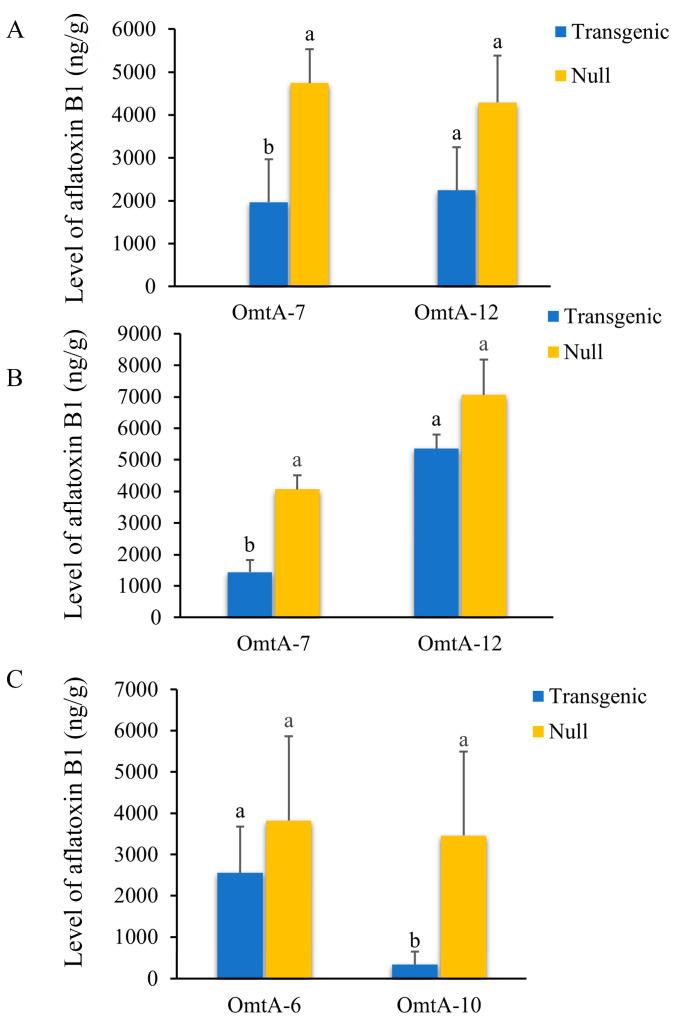
Evaluation of homozygous *omtA* transgenic lines under field conditions with artificial inoculations. The levels of aflatoxin in field-inoculated transgenic OmtA-7 and OmtA-12 lines, along with their null controls from the T5 generation in 2018 (**A**) and T6 generation in 2019 (**B**). Aflatoxin production in two additional transgenic lines (OmtA-6 and OmtA-10) and their corresponding null controls at the T4 generation was also determined in 2019 (**C**)**.** The inoculum concentration used for these experiments was 1 × 10^5^ conidia/mL. Different letters on top of the bars indicate a significant difference in aflatoxin B1 levels at *p* ≤ 0.05.

**Figure 5 toxins-17-00008-f005:**
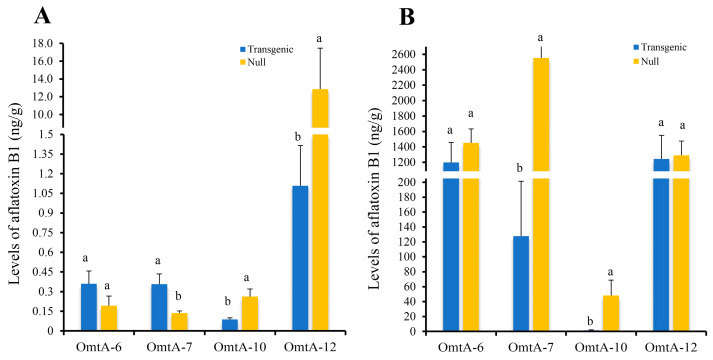
Aflatoxin production in surface-sterilized, field-grown mature kernels of homozygous omtA transgenic and null lines under kernel screening assay conditions. Aflatoxin production in the naturally infected mature transgenic kernels and their null counterparts harvested from the field from T4 (OmtA-6 and OmtA-10) and T6 (OmtA-7 and OmtA-12) generation without (**A**) or with (**B**) inoculation, followed by inoculation for 7 days at 31 °C under 100% humidity. Different letters on top of the bars indicate a significant difference in aflatoxin B1 levels at *p* ≤ 0.05. Fifteen kernels per line were used for the aflatoxin analysis.

**Figure 6 toxins-17-00008-f006:**
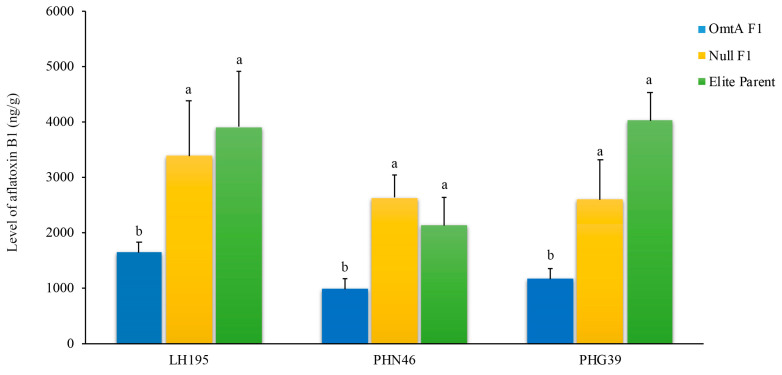
Aflatoxin B1 accumulation in the F1 crosses between OmtA-7 homozygous transgenic and elite inbred lines of LH195, PHN46, and PHG39, compared to crosses with null and self-pollinated elite lines under field inoculation conditions (*p* < 0.05). Different letters on top of the bars indicate a significant difference in aflatoxin B1 levels at *p* ≤ 0.05.

**Figure 7 toxins-17-00008-f007:**
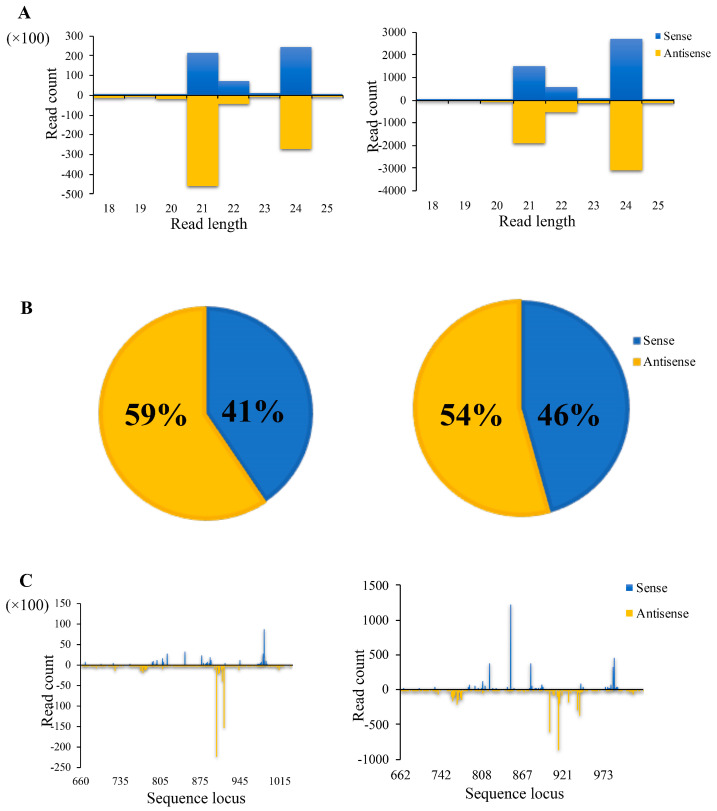
Distribution and frequency of small RNAs in leaf and immature kernel tissues of the transgenic maize line. (**A**). High levels of 21 and 24 nucleotide *omtA*-specific small RNAs were detected in the leaves of the transgenic OmtA-7 T3 generation (**left**) and in the kernels of the T4 generation (**right**), respectively. (**B**). A higher percentage of the anti-sense strand than the sense strand of *omtA*-specific small RNAs was detected in the transgenic maize line. (**C**). The 390 bp region of the *omtA* gene (between the 660 bp and 1015 bp) appears to be the hotspot for dsRNA production, with the highest number of gene-specific small RNA reads. The horizontal axis indicates the relative position of small RNAs aligned to the target *omtA* gene. The vertical axis represents the number of small RNA reads mapped to *omtA*.

**Table 1 toxins-17-00008-t001:** Chi-square analysis of the observed and expected T2 generation transgenic and non-transgenic seedlings to determine the *omtA* gene copy number in the four transgenic events.

Event	Number of Seedlings	Transgenic Seedlings (O/E) *	Non-Transgenic Seedlings (O/E) *	X^2^	# of Integration
OmtA-6	91	65/67	26/24	0.2	1
OmtA-7	83	64/65	23/22	0.1	1
OmtA-10	94	61/70	33/24	4.5	>1
OmtA-12	96	70/67	19/22	0.5	1

*: O, observed number of seedlings; E, expected number of seedlings. Whether the calculated chi-square Χ^2^ = ∑(observed − expected)2/(expected) exceeds the critical value (3.841, *p* < 0.05) for single-copy integration with an expected transgenic to null ratio of 3:1 is used to reject or accept the null hypothesis of being one or more than one copy.

**Table 2 toxins-17-00008-t002:** Zygosity estimation of purported homozygous T3 seedlings in comparison to the purported heterozygous T0 seedling based on real-time PCR quantification of the *omtA* gene and the reference alcohol dehydrogenase gene (*adh*).

Line	Heterozygous *	Homozygous	Ratio
OmtA-6	0.99	1.89	1:2
OmtA-7	1.0	2.0	1:2
OmtA-10	0.2	0.43	1:2
OmtA-12	0.6	1.1	1:2

* Zygosity = 2^(T3 Ct (*Ad*)−T3 Ct (*omtA*)/2(T0 Ct (*Adh*)−T0 Ct (*omtA*)^.

**Table 3 toxins-17-00008-t003:** Transgene copy number analysis of seedling leaf tissues collected from different *omtA* transformation events at T0 and T4 generations using droplet digital PCR.

Event	*OmtA* Gene (copy/µL)	*Adh1* Gene (copy/µL)	*OmtA*/*Adh1*	Copy Number *
OmtA-6 (T0)	626	156	4.01	4 (hemi)
OmtA-7 (T0)	176	187	0.94	1 (hemi)
OmtA-8 (T0)	238	245	0.97	1 (hemi)
OmtA-10 (T0)	247	261	0.95	1 (hemi)
OmtA-11 (T0)	196	200	0.98	1 (hemi)
OmtA-12 (T0)	211	240	0.88	1 (hemi)
OmtA-6 (T4)	769	96	7.9	4 (homo)
OmtA-7 (T4)	98.2	43.1	2.30	1 (homo)
OmtA-10 (T4)	133.4	63.0	2.12	1 (homo)
OmtA-12 (T4)	87	39	2.2	1 (homo)

* The ratio of the number of molecules of the *omtA* target gene vs. the number of molecules of the single-copy reference *Adh1* (alcohol dehydrogenase) gene is used to represent the copy number of the target gene in each of the transgenic events. There are two sets of binding sites in the HIGS-*OmtA* construct for the *omtA* primers used in the ddPCR.

**Table 4 toxins-17-00008-t004:** Phenotypic assessment of homozygous transgenic OmtA-RNAi lines in comparison with their null and B104 controls.

Line	Mean Plant Height (cm)	Days to Tasseling	Days to Silking	Mean Cob Length (cm)	Mean 100-Kernel Weight (g)
OmtA-6 (T)	132.5 e*	69.0 a*	71.0 b*	12.8 e*	20.9 d*
OmtA-6 (N)	133.3 de*	68.0 b	72.1 a*	13.9 cd	21.1 cd*
OmtA-7 (T)	137.4 bc	68.1 b	71.0 b*	14.4 c	23.2 b*
OmtA-7 (N)	138.8 ab	68.0 b	70.1 c	13.4 de	21.7 cd*
OmtA-10 (T)	139.4 ab	67.0 c*	70.0 c	13.7 cd	23.1 b*
OmtA-10 (N)	141.4 a	67.1 c*	70.0 c	15.1 bc	23.1 b*
OmtA-12 (T)	135.7 cd*	68.1 b	71.1 b*	15.5 a*	22.1 bc*
OmtA-12 (N)	134.3 de*	68.0 b	71.0 b*	16.2 a*	24.7 a
B104- (N)	139.4 ab	68.0 b	70.0 c	14.3 d	24.8 a
** Std. dev.	3.1	0.6	0.7	1.1	1.4

* Indicate lines with significant differences in the phenotypic trait compared to the B104 parental plants at *p* < 0.05. Lines followed by different letters are significantly different for the phenotypic trait at *p* < 0.05. ** Standard deviation for the phenotypic trait. Ten to thirteen representative plants/ears were used for each of these analyses.

**Table 5 toxins-17-00008-t005:** Summary of small RNA reads from the libraries of leaves at T3 generation and immature kernels at T4 generation of OmtA transgenic and non-transgenic maize lines.

Tissue Type	Event	Total Read	Reads Aligned to *A. flavus*	Reads Aligned to *omtA*
Leaf	OmtA-7 (Homo)	29,290,448	378,137	77,671
	B104 (Null)	60,780,742	6238	25
Kernel	OmtA-7 (Homo)	25,576,020	51,791	12,858
	OmtA-12 (Homo)	55,055,405	78,161	11,658
	OmtA-7 (Null)	35,595,011	52,163	13
	OmtA-12 (Null)	27,880,782	40,548	28

## Data Availability

The small RNA sequence datasets generated for this study have been deposited into the Sequence Read Archive (SRA) database (https://www.ncbi.nlm.nih.gov/sra/PRJNA771996) under the accession number PRJNA771996 created on 17 October 2021.

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
