# Peer review of "Host-Induced Gene Silencing of the Aspergillus flavus O-Methyl Transferase Gene Enhanced Maize Aflatoxin Resistance"

_toxins, 2024, doi:10.3390/toxins17010008_

Round 1
Reviewer 1 Report
Comments and Suggestions for Authors
Reviewer’s comments
1. The surface sterilization needs to be clarified. Could the authors specify how the surface sterilization process was carried out? Did the authors use any chemical agents for this purpose?
2. What type of growth media was used for inoculum cultivation in line 307?
3. In line 535, Please provide detailed information on the cDNA synthesis process
4. Line 538: Please explain the method used for performing serial dilution to evaluate primer amplification efficiency.
5. Recommended to provide the HPLC chromatogram for cross-verification of the aflatoxin quantification.
6. In Figure 1B, the Omt A -10 appears as multiple bands. Can you clarify if this represents distinct expression patterns or overlapping signals?
7. Please recheck and correct the typographical error in line 629. "UV right" should be corrected to "UV light."
8. Please recheck the reference formats
Comments on the Quality of English Language
I am not qualified to comment on the quality of the English language in the manuscript.
Author Response
Comments from Reviewer #1
- The surface sterilization needs to be clarified. Could the authors specify how the surface sterilization process was carried out? Did the authors use any chemical agents for this purpose?
Response: The surface sterilization of corn kernels was done through soaking the seeds with 70% ethanol for 5 minutes, followed by rinsing them with sterile water 3 times to remove any residual ethanol. We have added the details in the revised manuscript with reference
- What type of growth media was used for inoculum cultivation in line 307?
Response: The growth medium of Potato Dextrose Agar (PDA) was used for growing A. flavus to collect inoculum. This information was provided in the Materials and Methods section in Line 593-594 “A. flavus was cultured on PDA plates then diluted 1 x 105 conidia/ml in 0.01% SDS solution of freshly inoculum prepared from 7-day old A. flavus AF13 (ATCC 96044, SRRC 1273)”
- In line 535, Please provide detailed information on the cDNA synthesis process
Response: cDNA was synthesized from 250 ng of total RNA using the TaqMan reverse transcription reagents (Applied Biosystems, Foster City, CA, USA) according to the manufacturer’s recommendations. RNase-free DNase (Qiagen) was used to remove possible residual DNA contamination. We have added the relevant information in the revised manuscript.
- Line 538: Please explain the method used for performing serial dilution to evaluate primer amplification efficiency.
Response: The material has been added “Five-fold serial dilutions of cDNA or genomic DNA from 100 ng to 10 pg were used as templates in real time PCR to determine amplification efficiency” (see line 601 in the revised manuscript).
- Recommended to provide the HPLC chromatogram for cross-verification of the aflatoxin quantification.
Response: We always examine the HPLC chromatogram to verify the correct peak identification in aflatoxin quantification through HPLC. We have added a few HPLC chromatograms here (Figure S4) to relieve the concerns from the reviewer. However, I have rarely seen any papers publishing HPLC chromatogram for aflatoxin cross-verification purpose. Therefore, we decided to leave the figures/tables as they were.
- In Figure 1B, the Omt A -10 appears as multiple bands. Can you clarify if this represents distinct expression patterns or overlapping signals?
Response: I believe you mean Figure 1A. The second band (the lower and much faint one) is a band resulted from non-specific amplification from maize genomic DNA as you can see this band is absent in the positive control on the right when the plasmid was used as a template. It does not represent a distinct expression or overlapping signal.
- Please recheck and correct the typographical error in line 629. "UV right" should be corrected to "UV light."
Response: The correction has been made. Thanks for catching our mistake.
- Please recheck the reference formats
Response: The reference formats have been rechecked
Reviewer 2 Report
Comments and Suggestions for Authors
The authors studied an RNA interference (RNAi) approach called host-induced gene silencing (HIGS), which was used to suppress the O-methyl transferase gene (omtA, also called aflP), a key gene involved in aflatoxin biosynthesis. For this purpose, an RNAi vector carrying part of the omtA gene was introduced into the B104 maize line. Of the six transformation events that were positive for the omtA transgene content, OmtA-6 and OmtA-10 were selfed from T1 to T4 and OmtA-7 and OmtA-12 were generated at T6. These four lines, under laboratory conditions, showed a reduction of at least 81.3% in aflatoxin accumulation in the T3 generation, however, when tested in field conditions, with artificial inoculation of OmtA-7 in the T5 and T6 generations and OmtA-10 in the T4 generation, they showed a reduction between 60% and 91% in aflatoxin accumulation (P<0.04).
The abstract is well formulated in such a way that the results obtained and their degree of innovation are well presented.
The introduction analyzes 51 bibliographic references regarding the critical analysis of the current state of knowledge in the field of aflatoxin biosynthesis, which have been rigorously selected from the specialized literature.
In the two chapters: results and discussions, the authors use 5 tables and 7 very suggestive and well-edited figures to present the experimental data. The data obtained are rigorously discussed.
In the experimental part, a series of systems and analysis methods are well described, well presented in the sense of a gradual description in an open causal chain (Construction and transformation of maize of an RNAi vector to target the A. flavus omtA gene; Confirmation of transformation and expression of the target gene; Multiplication of selected transgenic lines and production of crosses with elite maize lines; Determination of the number of transgene copies transgenic maize lines for changes in aflatoxin accumulation in laboratory and field conditions; Extraction and quantification of aflatoxin; Total RNA isolation, construction of a small RNA library; Sequencing and bioinformatics detection of small RNA specific to the omtA gene and Statistical analyses), allowing a clear interpretation of the results obtained.
The conclusions are well formulated and well support the experimental results obtained by the authors.
89 references are selected from the specialized literature in the field, which in their presentation comply with the journal's norms.
I agree to the publication of the manuscript!
Author Response
Comments from Reviewer #2:
The authors studied an RNA interference (RNAi) approach called host-induced gene silencing (HIGS), which was used to suppress the O-methyl transferase gene (omtA, also called aflP), a key gene involved in aflatoxin biosynthesis. For this purpose, an RNAi vector carrying part of the omtA gene was introduced into the B104 maize line. Of the six transformation events that were positive for the omtA transgene content, OmtA-6 and OmtA-10 were selfed from T1 to T4 and OmtA-7 and OmtA-12 were generated at T6. These four lines, under laboratory conditions, showed a reduction of at least 81.3% in aflatoxin accumulation in the T3 generation, however, when tested in field conditions, with artificial inoculation of OmtA-7 in the T5 and T6 generations and OmtA-10 in the T4 generation, they showed a reduction between 60% and 91% in aflatoxin accumulation (P<0.04).
The abstract is well formulated in such a way that the results obtained and their degree of innovation are well presented. The introduction analyzes 51 bibliographic references regarding the critical analysis of the current state of knowledge in the field of aflatoxin biosynthesis, which have been rigorously selected from the specialized literature.
In the two chapters: results and discussions, the authors use 5 tables and 7 very suggestive and well-edited figures to present the experimental data. The data obtained are rigorously discussed.
In the experimental part, a series of systems and analysis methods are well described, well presented in the sense of a gradual description in an open causal chain (Construction and transformation of maize of an RNAi vector to target the A. flavus omtA gene; Confirmation of transformation and expression of the target gene; Multiplication of selected transgenic lines and production of crosses with elite maize lines; Determination of the number of transgene copies transgenic maize lines for changes in aflatoxin accumulation in laboratory and field conditions; Extraction and quantification of aflatoxin; Total RNA isolation, construction of a small RNA library; Sequencing and bioinformatics detection of small RNA specific to the omtA gene and Statistical analyses), allowing a clear interpretation of the results obtained.
The conclusions are well formulated and well support the experimental results obtained by the authors.
89 references are selected from the specialized literature in the field, which in their presentation comply with the journal's norms.
I agree to the publication of the manuscript!
Responses: There were no comments from Reviewer #2 for improvement during revision. We really appreciate the time and effort of the reviewer #2 invested in reviewing this manuscript.
Reviewer 3 Report
Comments and Suggestions for Authors
The topic is very interesting and within the scope of this Journal.
Abstract
Lines 14 and 19: I don’t understand why omt is italicized in some parts and not in others
Introduction
Line 45: Maybe it is better to remove “20 parts per billion (ppb)” because it is well written with 20 ng/g.
Line 47: Please, check and specify the European regulation for the aflatoxin in food and feed (6 ppb).
Line 68: In this sentence: “aflR, which encodes a transcription factor”, could you complete more about its functional impact?. When you talk about aflB and aflA you specify more.
If the goal is to study the omtA (aflP) gene further, I believe the research conducted on it should be better justified in the introduction.
I would like to see in the introduction somewhere what the general advantages of using the HIGS strategy are compared to others.
Results
Line 142: Somewhere in the results parts (line 521) you talk about Agrobacterium tumefaciens. Maybe it is better to homogenize and write A. tumefaciens here too.
Line 153: Figure 1. I don't see the bottom parts of the error bars in the figure.
Line 176: Maybe you have to specify what is the meaning of ddPCR.
Line 217: Maybe you have to write (specify) aflatoxin B1.
Line 229: A P value of 0.05 is considered significant, so clarifying this interpretation would be better for clarifying.
Line 231: Please it is better to write the units in values (ng/g for example) not in ppb. I don't see the bottom parts of the error bars in the figure.
Line 244: “There was a significant delay” Please, specify.
Line 267: Please, change ppb for ng/g.
Line 300: Figure 4. I don't see the bottom parts of the error bars in the figure. Please, change ppb for ng/g.
Line 322. Figure 5. I don't see the bottom parts of the error bars in the figure. Please, change ppb for ng/g.
Line 331. Figure 5. I don't see the bottom parts of the error bars in the figure. Please, change ppb for ng/g.
Discussion
Please change ppb for ng/g in this part.
Materials and methods
Line 515: I don’t understand well this sentence: The homologous recombination to produce the final pBS-d35S-attB4-5' OmtA-attB1-PR10 intron-CmR-attB2-3’ OmtA-attB3 was performed as previously described [51]. What does it mean? Described previously by the same authors?.
Line 564: “The presence or absence of the target gene omtA as described previously [49]”. What does it mean? Described previously by the same authors?.
Line 615: Why don't you analyze aflatoxins B2, G1, and G2? The title is misleading because it is not specific to aflatoxin B1.
Line 617: “80% methanol in 50-mL flasks as previously described [49].” What does it mean? Described previously by the same authors?.
Line 618: Whatman paper? I don´t know if Waterman filter is a mistake.
Line 633: Could you please specify the points of the standard curve?
References:
In some parts, the DOI appears, and in others, it does not. Please check all the references.
Line 700: Please delete “pp” for the pages.
Line 713: What is the meaning of “e45151”?
Line 793: What is the meaning of “e1602382”?
Line 827: What is the meaning of “e126185”?
Author Response
Comments from Reviewer #3:
The topic is very interesting and within the scope of this Journal.
Abstract
Lines 14 and 19: I don’t understand why omt is italicized in some parts and not in others.
Response: We use “omtA” in italic lower case to represent the gene and the non-italicized “OmtA” followed by a number to represent different RNAi-omtA transformation events.
Introduction
Line 45: Maybe it is better to remove “20 parts per billion (ppb)” because it is well written with 20 ng/g.
Response: The suggested change has been made.
Line 47: Please, check and specify the European regulation for the aflatoxin in food and feed (6 ppb).
Response: The latest EU standards issued in May 8 2023 on maximum levels of of Aflatoxins in food and feed are quite complex depending on the crop and the intended use, ranging from 2.0 ng/g of B1 in cereals and dried fruits to 12 ng/g for almonds, pistachios and apricot kernels to be subjected to sorting or other physical treatment before placing on the market. The limit on maize and rice is 5 ng/g to be subjected to sorting or other physical treatment before placing on the market. We have revised this part accordingly.
Line 68: In this sentence: “aflR, which encodes a transcription factor”, could you complete more about its functional impact?. When you talk about aflB and aflA you specify more.
Response: we have added a reference on the function and importance of aflR in the revised manuscript.
If the goal is to study the omtA (aflP) gene further, I believe the research conducted on it should be better justified in the introduction.
Response: the goal of this paper is to investigate whether suppressing the omtA expression through HIGS can reduce aflatoxin contamination in transgenic maize, not to study the function of omtA further. This information was present in the original manuscript (second to the last paragraph in the introduction)
I would like to see in the introduction somewhere what the general advantages of using the HIGS strategy are compared to others.
Response: Thanks for your suggestion. We have added the information in the last paragraph of the discussion section in the revised manuscript.
Results
Line 142: Somewhere in the results parts (line 521) you talk about Agrobacterium tumefaciens. Maybe it is better to homogenize and write A. tumefaciens here too.
Response: Follow your suggestion, we used the full name here since it was the first time this species is mentioned in the paper. We did change the full name of “Agrobacterium tumefaciens” on line 521 to “A. tumefaciens” in the revised manuscript.
Line 153: Figure 1. I don't see the bottom parts of the error bars in the figure.
Response: We intended not to show the bottom half of the error bars. In this way, the bars look more neat or tidy. Also, when the bars are too short, the bottom half of the error bars would extend to below the x-axis if they are shown.
Line 176: Maybe you have to specify what is the meaning of ddPCR.
Response: ddPCR stands for “droplet digital PCR”. The full name was used at Line 171 of the original version of manuscript when it was first used.
Line 217: Maybe you have to write (specify) aflatoxin B1.
Response: Thanks very much for your suggestion and we have made it clear by adding “B1” after aflatoxin.
Line 229: A P value of 0.05 is considered significant, so clarifying this interpretation would be better for clarifying.
Response: P value refers to is a statistical probability, which means there is 95% of chance or confidence that two values in question are significantly different when they are indicated with different letters.
Line 231: Please it is better to write the units in values (ng/g for example) not in ppb. I don't see the bottom parts of the error bars in the figure.
Response: We have changed the ppb to ng/g. Regarding the bottom half of the error bars, please see our response to your comment for Line 153.
Line 244: “There was a significant delay” Please, specify.
Response: It means those plants flowering time is delayed compared to B104 and this delay is statistically significant.
Line 267: Please, change ppb for ng/g.
Response: done.
Line 300: Figure 4. I don't see the bottom parts of the error bars in the figure. Please, change ppb for ng/g.
Response: Please see the reasons in response to your comment for Line 153. We have changed the ppb to ng/g in the revised Figure 4.
Line 322. Figure 5. I don't see the bottom parts of the error bars in the figure. Please, change ppb for ng/g.
Response: See above. We have changed ppb to ng/g in revised Figure 5.
Line 331. Figure 5. I don't see the bottom parts of the error bars in the figure. Please, change ppb for ng/g.
Response: See above. We have changed ppb to ng/g in revised Figure 5.
Discussion
Please change ppb for ng/g in this part.
Response: done
Materials and methods
Line 515: I don’t understand well this sentence: The homologous recombination to produce the final pBS-d35S-attB4-5' OmtA-attB1-PR10 intron-CmR-attB2-3’ OmtA-attB3 was performed as previously described [51]. What does it mean? Described previously by the same authors?.
Response: the base vector used for constructing the HIGS was made in previous study by the corresponding author Chen et al. (51) using a Gateway system that joins different DNA fragments in a specific order in one recombination reaction. This order is controlled by the sequences added to the ends of the fragment (attB/attL).
Line 564: “The presence or absence of the target gene omtA as described previously [49]”. What does it mean? Described previously by the same authors?.
Response: It means we used the same PCR method that was used in our previous study by the same authors Omolehin et al. (49) to determine whether the leaf tissue from each of the transformation events contains the target omtA gene or not.
Line 615: Why don't you analyze aflatoxins B2, G1, and G2? The title is misleading because it is not specific to aflatoxin B1.
Response: We did analyze the levels of all four aflatoxins in all the samples including B1, B2, G1 and G2. However, B2, G1 and G2 are proportionally much lower. Therefore, we only report B1 levels and it has been a good and reliable indicator of aflatoxin resistance. Therefore, the title is not misleading at all.
Line 617: “80% methanol in 50-mL flasks as previously described [49].” What does it mean? Described previously by the same authors?.
Response: It means we used the same method to extract aflatoxin from ground kernels as described in our previous publication by the same authors Omolehin et al. (49).
Line 618: Whatman paper? I don´t know if Waterman filter is a mistake.
Response: what we have in the original manuscript “No. 1 Waterman filter paper” is correct. Whatman is a brand of filter papers.
Line 633: Could you please specify the points of the standard curve?
Response: our aflatoxin standard curve has 7 data points: 1, 5, 10, 50, 100, 500, and 1000 ng/g (ng/ml). We have added this information in the revised manuscript.
References:
In some parts, the DOI appears, and in others, it does not. Please check all the references.
Response: DOI refers to Digital Objective Identifier. It provides a permanent web address for some recent published papers to help the readers to access them quickly through the link. For papers published a while back, there are no DOI available. Therefore, they are not provided.
Line 700: Please delete “pp” for the pages.
Response: What we have in the manuscript is correct. The reason the pp is there is to indicate the cited reference is a book chapter, not a journal, which we do not use pp to specify page ranges.
Line 713: What is the meaning of “e45151”?
Response: When the publication is in electronic form only, there is no paper number, instead, it uses an “e” followed by a specific number. What we had there in the reference is correct.
Line 793: What is the meaning of “e1602382”?
See response above
Line 827: What is the meaning of “e126185”?
See response to line 713.
Round 2
Reviewer 1 Report
Comments and Suggestions for Authors
The authors satisfactorily addressed all my comments and it can be accepted for publication in the present form.